# Enhanced Wettability, Hardness, and Tunable Optical Properties of SiC_x_N_y_ Coatings Formed by Reactive Magnetron Sputtering

**DOI:** 10.3390/ma16041467

**Published:** 2023-02-09

**Authors:** Veronica S. Sulyaeva, Alexey N. Kolodin, Maxim N. Khomyakov, Alexander K. Kozhevnikov, Marina L. Kosinova

**Affiliations:** 1Nikolaev Institute of Inorganic Chemistry SB RAS, 3, Acad. Lavrentiev Ave., 630090 Novosibirsk, Russia; 2Institute of Laser Physics SB RAS, 15B, Acad. Lavrentiev Ave., 630090 Novosibirsk, Russia

**Keywords:** silicon carbonitride films, magnetron sputtering, hydrophilic films, tunable optical properties, mechanical properties, film aging

## Abstract

Silicon carbonitride films were deposited on Si (100), Ge (111), and fused silica substrates through the reactive magnetron sputtering of a SiC target in an argon-nitrogen mixture. The deposition was carried out at room temperature and 300 °C and at an RF target power of 50–150 W. An increase in the nitrogen flow rate leads to the formation of bonds between silicon and carbon atoms and nitrogen atoms and to the formation of SiC_x_N_y_ layers. The as-deposited films were analyzed with respect to their element composition, state of chemical bonding, mechanical and optical properties, and wetting behavior. It was found that all synthesized films were amorphous and represented a mixture of SiC_x_N_y_ with free carbon. The films’ surfaces were smooth and uniform, with a roughness of about 0.2 nm. Depending on the deposition conditions, SiC_x_N_y_ films within the composition range 24.1 < Si < 44.0 at.%, 22.4 < C < 56.1 at.%, and 1.6 < N < 51.9 at.% were prepared. The contact angle values vary from 37° to 67°, the hardness values range from 16.2 to 34.4 GPa, and the optical band gap energy changes from 1.81 to 2.53 eV depending on the synthesis conditions of the SiC_x_N_y_ layers. Particular attention was paid to the study of the stability of the elemental composition of the samples over time, which showed the invariance of the composition of the SiC_x_N_y_ films for five months.

## 1. Introduction

Owing to their multifaceted properties, silicon carbonitrides SiC_x_N_y_ [1,2,3] are modern multifunctional materials with a wide range of applications. SiC_x_N_y_ exhibits a combination of tribo-mechanical (high hardness [4,5,6,7,8,9,10,11,12,13,14,15,16,17], wear resistance [6,7], low coefficient of friction [6], high adhesion to construction materials [6]), chemical (high thermal and corrosion resistance [1,7,10]), and physical (low dielectric constant [1], controlled in a wide range of values of refractive index [18] and optical band gap [18,19,20,21], high optical transparency in a wide spectral range [19,21,22,23], high hydrophobicity [24]) properties. Due to the variable composition of the compound, it is possible to change the values of the conductivity, the hardness, and Young’s modulus. The careful design of the composition of SiC_x_N_y_ films allows band gap engineering, tuning of the light absorption and refractive index. The multipurpose orientation of the use of these films, the expansion of functionality, and the optimization of characteristics constantly put forward new tasks for researchers. This explains the continuing relevance of multifaceted studies of film deposition processes after more than forty years. SiC_x_N_y_ film and coating synthetic routes are based on both chemical and physical deposition methods [1,3]. Among the physical methods, magnetron sputtering (MS) is the most frequently used technique, and its various types include radio frequency (RF) sputtering, direct current (DC) sputtering, and high-power impulse magnetron sputtering (HiPIMS). It is important to emphasize that, in contrast to CVD processes, magnetron sputtering can produce films without hydrogen-containing bonds. Since magnetron sputtering is one of the most versatile methods for depositing films of good quality at low temperatures, and this process is easily scalable, we consider in more detail the variations in the deposition conditions in such processes and the functional properties of the obtained layers. For the synthesis of SiC_x_N_y_ films, over the course of many years of research, various combinations of target types and reaction gases were tested. Table 1 presents a generalization of the experimental data on SiC_x_N_y_ film deposition using MS processes.

SiC, Si, and C (graphite) are most often used as targets in magnetron sputtering experiments. Some studies have been carried out with Si_3_N_4_ [23] and segmented Si+C targets [25,26]. The power applied to the targets in conventional RF and DC MS processes varies over a wide range of 80 to 600 W, while, in the HiPIMS process, it is much higher (up to 1400 W). In many cases, the formation of SiC_x_N_y_ films is achieved by the reactive sputtering of these targets in argon-nitrogen media. The sputtering of a single silicon target is accompanied by the presence of a hydrocarbon gas such as CH_4_ or C_2_H_2_. The authors of the works mentioned in the Table 1 showed the possibility of varying the composition of SiC_x_N_y_ films over a wide range through the use of different amounts of reactive gases (N_2_, CH_4_ or C_2_H_2_) during sputtering. It is known that the addition of gaseous nitrogen to the reaction chamber is an effective method for the dosed introduction of nitrogen into the film composition. As can be seen from Table 1, the most common reactive gas used in the sputtering processes is nitrogen. The introduction of N_2_ into the gas mixture leads to an increase in the nitrogen content of the SiC_x_N_y_ films of up to 40 at.% [9]. A similar effect of the addition of nitrogen-containing gas on the composition of SiC_x_N_y_ films is also shown for CVD films [1]. As follows from the literature, most of the films were obtained at room temperature (RT). There is a very limited number of works in which the authors have used elevated deposition temperatures of up to 700 °C [23]. Different authors used different experimental setups, target materials, target–substrate distances, levels of applied power, reactor chamber geometries, reactive gases, and flow rates; therefore, all of the above leads not only to difficulties in comparison, but to the potential for the comparison of results not to be scientifically perfect. Nevertheless, it is interesting to draw at least some correlations. To compare the influence of various magnetron sputtering processes and types of sputtered targets on the SiC_x_N_y_ films’ growth rates, we summarized the results reported in the literature in Figure 1. Considering that the growth rate depends critically on the target–substrate distance, it would be useful to add another axis with these data. Unfortunately, the target–substrate distance is almost impossible to deduce in a large number of articles (Table 2). It should be noted that, in most studies, the growth rate does not exceed 30 nm/min. A high growth rate of up to 80 nm/min was obtained during the RF magnetron sputtering of an SiC target, where the target–substrate distance was 25 mm [22].

Typically, the as-deposited SiC_x_N_y_ films produced in MS processes carried out at room temperature were amorphous. However, the work [13] showed that, although the films are amorphous, according to the XRD and TEM data, crystallites of β-Si_3_N_4_ of 5 nm in size were found. A limited number of works dealt with the study of film roughness. Depending on the conditions in which the films were obtained, the value of RMS varies in a very wide range from 0.2 to 65 nm [5,14]. It should be emphasized that, in most cases, SiC_x_N_y_ films have a very low roughness (Table 1), which is typical for amorphous films. The authors of [14] found that the films become smoother as the nitrogen fraction in the Ar + N_2_ mixture increases.

As SiC_x_N_y_ films are considered to be promising materials for high-temperature protective applications [41,42], special attention in the study of their functional properties was paid to the study of their mechanical performances. The authors studied the effects of the films’ growth parameters on the changes in their mechanical properties, such as hardness, elastic modulus, friction coefficient, and strength, while they rarely paid attention to determining the dependence of these properties on the elemental composition. The authors agree that changes in the composition and chemical structure of the films caused by variations in the parameters of the magnetron sputtering process (magnetron sputtering mode, composition of the gas mixture, target sputtering power, deposition temperature) lead to changes in the mechanical properties of the films. According to the data from the literature (Table 1), the films exhibited hardness (H) and elastic modulus (E) values with wide ranges, from 5 to 29 GPa and from 50 to 293 GPa, respectively. Only one group of researchers [13] studied such an important indicator of mechanical action as the elastic recovery (R) of SiC_x_N_y_ films.

Despite numerous studies of the mechanical characteristics (H and E) of films in the literature, there are contradictory data on the effect of the PVD process parameters and the film characteristics. According to some authors, the hardness values of the films decreased with an increasing content of nitrogen, specifically from H = 23 GPa for the pure SiC film to H = 18 GPa for the SiCN film with the highest nitrogen content (40 at.% of nitrogen) [7], from 22.5 GPa to 18.5 GPa (40 at.%) [9], and from 23 GPa to 17 GPa (22 at.%) [8]. In contrast, the authors of [4] determined that the films with the highest nitrogen content (38 at.%) had the maximum hardness (22 GPa). Its value remains almost constant up to 30 at.% nitrogen, and then drops sharply. The temperature dependence of hardness is more complex. In [16], the hardness increased from 10 to 22 GPa with a rise in the deposition temperature from room temperature to 500 °C, then dropped significantly to 9 GPa at 600 °C. These authors also considered the effect of the sputtering power of the SiC target at a film deposition temperature of 500 °C and found that an increase in power from 200 to 400 W led to an increase in hardness from 9 to 22 GPa. Meanwhile, in [17], directly the opposite relation was observed. With an increase in the RF sputtering power of the SiC target from 250 to 620 W, the hardness of the SiC_x_N_y_ films decreased monotonically from 15.5 to 14.2 GPa, while, in the case of the sputtering of the Si_3_N_4_ target, the hardness increased from 11.8 to 14.0 GPa as the power increased from 150 to 620 W. Thus, different tendencies are observed when using silicon carbide and silicon nitride targets. As found in the works [32,41], a comparison of the hardness of the MS SiC_x_N_y_ films obtained in the RF and DC modes showed that RF sputtering leads to the production of harder coatings.

Table 2 presents some data on the maximum values of hardness and Young’s modulus and the corresponding elemental compositions and types of chemical bonding in as-deposited films.

According to the XRD data, all of the films considered in Table 2 were amorphous and contained carbon as a separate phase, following the Raman spectroscopy data [42]. An exception is work [32], where the high-temperature films were an amorphous matrix containing β-Si_3_N_4_, C_3_N_4_ crystallites (RF mode) and Si, β-SiC crystallites (DC mode). As can be seen from Table 2, the hard films have different elemental and phase compositions and different chemical structures. The authors of [42] described the SiC_x_N_y_ films with a composition in the middle of the tie line SiC to Si_3_N_4_ of a Si–C–N triangle as exhibiting values of hardness of around 28 GPa. It is known that the mechanical properties of materials depend on the nature of the bonding among their constituent atoms [43]. The authors [8] reported that a reduction in the number of Si–C bonds and an increase in the number of Si–N and C–N bonds in the films cause a decrease in the hardness and the elastic modulus. However, in works [5,6], it was found that a higher N/Si ratio as well as a greater number of Si–N bonds and a smaller number of Si–Si bonds lead to an increase in hardness. A limited number of works dealt with the study of the relationship between mechanical properties and residual stress in MS SiC_x_N_y_ films. According to [4], the lower residual stress in the C-rich SiC_x_N_y_ films might contribute to their lower H and E values.

It is known from data in the literature that, using the post-growth treatment of the films, the hardness value can be increased from 18–22 to 21–32 GPa [7] and from 12.5–17.5 to 34 GPa [15] through annealing and pulsed electron beam irradiation, respectively. In the first case, the authors explain the change in the film properties by an increase in their atomic short-range ordering. In the second case, it is noted that the irradiation leads to the formation of free silicon in the film. Comparatively few studies have been made regarding the behavior of the optical properties of MS SiC_x_N_y_ films [18]. Table 1 shows the value ranges of such optical characteristics as the refractive index (n), transmittance with an indication of the wavelength region, and the optical band gap (*E_g_*). All studies emphasize a decrease in *n* and a simultaneous increase in *E_g_* with an increase in nitrogen content, and no conflicting results are observed.

There is only one work where the wettability of MS films has been studied [12]. The SiC_x_N_y_ layers prepared by the DCMS of silicon and carbon targets in a nitrogen-argon atmosphere in the temperature range of 100 to 700 °C exhibited contact angles within the interval of 57 to 87°, while CVD SiC_x_N_y_ films usually show a hydrophobic nature (CA = 96−161°) [44].

Moreover, analyzing Table 1, one can note the following: there are nearly no systematic studies of the temperature dependence of the films’ properties, the micrometer coatings are mainly deposited, and few works were devoted to the thin films. Regarding the functional characteristics of the obtained layers, generally, the focus was on the study of their mechanical properties. Based on the set of the considered characteristics of the deposited SiC_x_N_y_ films, we can generalize that the synthesis parameters open up the possibility of tailoring the film properties and functional characteristics, such as its mechanical and optical properties, based on the needs of the desired application. Motivated by the above, the objective of the present work is to explore the effect of nitrogen introduction, RF power, and deposition temperature on the characteristics of SiC_x_N_y_ thin films prepared by RF magnetron sputtering using a SiC target in an Ar–N_2_ atmosphere in order to improve their functional performances, such as their optical and mechanical performances. Particular emphasis is placed on elucidating the influence of the film formation parameters on the wettability and stability of the elemental composition of a-SiC_x_N_y_ films at storage.

## 2. Experimental Section

### 2.1. Materials

For film deposition, we used the following substrates: n-type Si(100) wafers (10 × 10 × 0.5 mm), Ge(111) wafers (10 × 10 × 1 mm), and fused silica plates (10 × 5 × 1 mm). Si(100) substrates were used for scanning electron microscopy (SEM), atomic force microscopy (AFM), Fourier transform infrared spectroscopy (FTIR), X-ray photoelectron spectroscopy (XPS), ellipsometry, and nanoindentation, scratch test, and wettability investigations. Ge(111) substrates were used for the X-ray energy dispersive spectroscopy (EDX) analysis. Fused silica plates were used for the UV-Vis transmittance study.

Before the film deposition process, the substrates were subjected to cleaning, including the stages of degreasing and chemical etching. All types of substrates were degreased using trichloroethylene and acetone (high purity grade). The Si(100) substrates were chemically treated sequentially in ammonia-peroxide, hydrochloric-peroxide etchants, and an HF solution. Ge(111) substrates were treated in a solution of HNO_3_ + HF + CH_3_COOH (3:4.5:9). Each stage was completed by washing with deionized water. Finally, the substrates were dried in a nitrogen flow at 50 °C. The substrates were then immediately placed inside a vacuum chamber that was evacuated.

The SiC target (99.9% purity) was used as a cathode for the magnetron sputtering process. Argon (high purity grade, 99.999 vol.%) and nitrogen (high purity grade, 99.999 vol.%) were used.

### 2.2. Film Growth

SiC_x_N_y_ coatings were formed using a reactive RF magnetron sputtering technique. The deposition system has been described in detail elsewhere [45]. An SiC target of 51 mm in diameter and 6 mm in thickness was sputtered at powers of 50–150 W. The target–substrate distance was 30 mm. The target was not pre-sputtered. The vacuum chamber was evacuated to a base pressure of 5 × 10^–5^ Torr. Argon was used as the sputtering gas, while nitrogen was used as the reactive gas for the deposition of nitrogen-containing films. We studied the effects of the following parameters on the properties of the films: the flow rates of the argon and nitrogen gases, the plasma power, and the substrate temperature. Three series of experiments were carried out by varying one parameter while keeping the others constant. Table 3 summarizes the experimental parameters under which the samples were obtained. In series A, the nitrogen flow rate was varied from 0 to 30 sccm in 5 sccm steps, while the total Ar + N_2_ flow in all experiments was maintained equal to 60 sccm. The working pressure in the reactor chamber was 4 × 10^–3^ Torr. The contribution of nitrogen consumption to film growth does not affect the total pressure in the system. The A and B series of SiC_x_N_y_ samples were synthesized at room temperature. The influence of the sputtering power of the target was studied in the B series. The samples in the C series were deposited at 300 °C with variation of the N_2_ flow rate. It is possible to achieve simultaneous deposition on three substrates with a 5% accuracy of thickness.

### 2.3. Film Stability Study

Two samples obtained by sputtering an SiC target in argon (F(Ar) = 60 sccm) and in a mixture of argon and nitrogen (F(Ar) = 30 sccm and F(N_2_) = 30 sccm) at room temperature and at 300 °C were chosen to study the stability of their elemental composition during storage in air. All other parameters of the deposition process were the same: the plasma power was 150 W, and the pressure in reactor chamber was 4 × 10^-3^ Torr. The aging of the films was examined by exposing the samples to air environments. The samples were stored in cleaned plastic Petri dishes under standard conditions. The elemental composition of the samples was analyzed immediately after synthesis, and then RT films were measured every day for two weeks, every week for two months, and then every month. In total, 20 measurements were carried out for each sample over the course of 5 months. The elemental composition of the films prepared at 300 °C was quantified promptly after their removal from the vacuum chamber and again after 3 months of air storage.

### 2.4. Characterization Techniques

The surface morphology of the films and their elemental compositions were studied using field-emission scanning electron microscope JEOL JSM 6700F (Jeol, Tokyo, Japan) equipped with Quantax 200 (Bruker, Berlin, Germany) analyzer for X-ray energy dispersive spectroscopy using an acceleration voltage of 5 keV according to [46]. The film thickness was determined via observing a cross-section view of the sample and also by ellipsometry.

The thickness and refractive index of the films were calculated using data from variable angle monochromatic null ellipsometry (He-Ne laser, λ = 632.8 nm). The measurements were carried out at 5 incidence angles. The film growth rate was calculated by dividing the film’s thickness by the deposition time.

The film surface topography was observed through atomic force microscopy using a Multimode 8 AFM (Bruker, Berlin, Germany) operating in soft-tapping mode with silicon cantilevers under ambient conditions. The analysis of the roughness surface profiles was performed using Gwyddion SPM data processing software.

The information on chemical bonding was acquired using FTIR and X-ray photoelectron spectroscopy. The Fourier transform spectrometer SCIMITAR FTS 2000 (Digilab, Hopkinton, MA, USA) was used for obtaining the FTIR absorption spectra of the layers in the region of 400–4000 cm^–1^ with a resolution of 2 cm^–1^. To control the content of the carbon phase in films, Raman spectra were recorded on a LabRAM HR Evolution (Horiba, Kyoto, Japan) spectrometer (He-Ne laser, λ = 514.5 nm).

The surface elemental composition of the SiC_x_N_y_ films was examined according to the XPS method using an X-ray photoelectron spectrometer (SPECS Surface Nano Analysis GmbH, Berlin, Germany). The core-level spectra were recorded using non-monochromatic Al K_α_ radiation with hν = 1486.6 eV. The C1s peak at 284.5 eV corresponding to surface adventitious carbon contamination was used for the charge correction of the XPS binding energy scale. The relative concentrations of elements were determined from the integrated intensities of the core-level spectra using the cross sections according to Scofield [47]. For detailed analysis, the spectra were fitted into several peaks using a Shirley background. The fitting procedure was performed using CasaXPS 2.3.24PR1.0 software. The line shape of the peaks was approximated by the multiplication of the Gaussian and Lorentzian functions.

The study of the sample structure was performed on a Shimadzu XRD-7000 diffractometer (Shimadzu, Kyoto, Japan) using CuK_α_ radiation (λ = 0.15456 nm, Ni filter) in the 2θ range of 5–60° (length step of 0.03°).

The UV-Vis-NIR transmittance of the SiC_x_N_y_/SiO_2_ structures was measured using a Shimadzu UV-3101PC scanning spectrophotometer (Shimadzu, Kyoto, Japan) in the range of 190–3200 nm with a resolution of 2 nm.

The mechanical properties, such as the hardness (H), elastic modulus €, and elastic recovery (R) of the coatings, were defined by nanoindentation measurements according to ISO 14577 with the scanning nano-hardness tester NanoScan-3D (TISNCM, Troitsk, Moscow, Russia) at several loads in the range from 1 to 70 mN, with at least 10 indents for each load to reduce the effect of the random error. Thus, the dependence of the effective hardness of the “film-substrate” system on the indenter penetration depth was analyzed. The obtained dependence was approximated using the Korsunsky’s model [48] for averaging the experimental data and determining the hardness of the film without the influence of the substrate. The adhesive strength of the coatings was evaluated through a variable load scratch test. For this, two scratches of the edge and face of the Berkovich diamond indenter (TISNCM, Troitsk, Moscow, Russia) were applied to each sample with a linearly rising normal load from 0.1 to 100 mN.

The measurements of the contact angles of the films were carried out on an OCA 15 PRO instrument (Dataphysics, Filderstadt, Germany), equipped with a measuring video system with a USB camera, as well as a fast-measuring lens with an adjustable viewing angle. All measurements were performed under normal conditions in a thermostatic box (T = 25 ± 2 °C and p = 750 Torr). The diameter of the needle was 0.51 mm. Distilled water was used as a test fluid. The droplet volume was constant and amounted to ~2.0 μl. The values of the contact angles were measured in the regime of a sessile drop. The calculation of the observed contact angle was carried out according to two algorithms: Ellipse-Fitting and the Young–Laplace algorithm. The final values of the contact angles were calculated as the average of 3 measurements.

## 3. Results and Discussion

The influence of the deposition temperature (T_dep_) and the N_2_ fraction in the argon-nitrogen mixture on the surface morphology, growth rate (V), elemental composition, chemical bonding, mechanical and optical properties, and wettability of the a-SiC_x_N_y_ layers are studied.

### 3.1. Film Growth Rate

Layers with a thickness from 140 nm to 320 nm were obtained. The film growth rate has been calculated by dividing the film thickness (SEM data and ellipsometry measurements) by the sputtering time. The deposition rate was investigated with respect to its dependence on the nitrogen flow rate (Figure 2a) and the RF power (Figure 2b).

When nitrogen is added from 0 to 10 sccm, the deposition rate increases from 78 to 103 nm/min, and it slightly decreased upon a further increase in F(N_2_). This phenomenon can be explained due to the formation of a chemical compound with a lower sputtering yield on the target surface. It is typical for reactive magnetron sputtering processes [49]. In our case, nitridization on the surface of the SiC target is possible. The thickness of this layer and the poisoned fraction surface of the target increase as the nitrogen flow rate increases. The study of the film growth rate during target sputtering into an argon-nitrogen mixture was carried out in works [22,34]. However, the authors found a monotonic decrease in the growth rate with the increase in the N_2_/Ar ratio. It should be noted that, in these experiments, the total flow of argon with nitrogen was not kept constant, which could lead to a change in the total pressure in the reaction chamber. Figure 2b depicts the effect of sputtering power on the kinetic characteristics of the film deposition process. As expected, the film growth rate increases with the increase in power from 50 to 150 W (series B). This dependence agrees with information from the literature for the same processes [38]. As can be seen from Figure 1, the obtained film deposition rates are higher than those previously reported in the literature. One of the reasons for obtaining high film deposition rates is the shorter target–substrate distance in our sputtering system (Table 2). No attention has been paid in the above-mentioned papers (Table 1) to studying the dependance of the growth rate of SiC_x_N_y_ on this parameter. However, for other systems, it was found that reducing the distance from 90 to 45 mm and from 70 to 35 mm leads to an increase in the film deposition rate of about 4 and 9 times, respectively [50,51]. According to the authors of [50], decreasing the target-to-substrate distance increases the probability of depositing more atoms onto the substrate in a shorter period of time, leading to the high deposition rates.

### 3.2. Film Surface Morphology, Topology and Structure Study

The surface morphologies of SiC_x_N_y_ films grown on Si(100) substrates were characterized using SEM. The films’ surfaces were a very smooth, without any notable features (Figure 3a,b).

The cross-sectional SEM image of the SiC_x_N_y_/Si structure displayed a flat interface between the film and the substrate (Figure 3c,d). Surface roughness measurements of SiC_x_N_y_ films were determined using the AFM. The root mean square (RMS) roughness was calculated using AFM scans with areas of 5 µm × 5 µm. The AFM planar view of the SiC_x_N_y_ films showed that these samples had a smooth and homogeneous surface topology. The RMS roughness value of these samples was equal to 0.2 nm and was independent of the value of the nitrogen flow. Two-dimensional AFM images of the SiC_x_N_y_ layers are shown in Figure 3e,f.

The X-ray diffraction patterns of the SiC_x_N_y_ samples did not contain any reflexes, which indicates the amorphous nature of the deposited films (Appendix A). There were no changes in the surface morphology and phase composition of films with the increase in the synthesis temperature to 300 °C (series C).

### 3.3. Chemical Composition of SiC_x_N_y_ Films

#### 3.3.1. EDX Data

An energy dispersion analysis was performed for the samples deposited on Ge(111) substrates to exclude the effect of the characteristic radiation of the substrate on the determination of the Si element concentration. The EDX spectra confirmed the presence of silicon, carbon, nitrogen, and a small amount of oxygen. In all of the energy dispersion spectra of the synthesized layers, a small peak of argon can also be observed, the content of which in the samples does not exceed 1 at.%, which is natural for films obtained using similar magnetron sputtering systems [41]. When nitrogen is added to the reaction mixture, its content in the film increases, while silicon and carbon decrease with the same tendency (Figure 4a), and their ratio is approximately equal to one, as in the initial target of silicon carbide.

A similar trend is observed during the deposition of SiC_x_N_y_ films obtained by the sputtering of a SiC target onto a substrate heated to 300 °C (series C, Table 4).

Thus, the addition of nitrogen provided elemental composition tunability from a silicon carbide-like film to silicon carbonitrides. It should be noted that the nitrogen content in films close in composition to SiC obtained by the sputtering of target in an Ar medium (0% N_2_ in the gas mixture) is of the order of 1–5 at.%, which is probably a consequence of the inclusion of nitrogen from the residual atmosphere in the vacuum chamber during the experiment. It is important to note that oxygen was not provided during the deposition and, despite that, its presence is not negligible (between 3 and 15 at.%), so we assumed that a certain amount of oxygen was included in the film during the growth process from residual atmosphere.

The aging of the samples during their storage in the air atmosphere under standard conditions was studied. The SiC_x_N_y_ films with the most different nitrogen contents (4 and 49 at.%, series A) were selected as the samples to be studied. The elemental composition of these samples was measured over the course of five months. Figure 4b,c shows that the SiC_x_N_y_ films retained the constancy of their elemental compositions, taking into account the measurement error by the EDX method (~10%), over the entire period of time, regardless of their nitrogen content. The oxygen content in the as-grown low- and high-nitrogen films was 4.1 and 4.8 at.%, respectively. The average amount of the oxygen content during storage was 4.2 and 5.5 at.% for the above-mentioned films. Taking into account the data on the stability of the RT films, we expected that the high-temperature films (series C) would also be stable, so the elemental composition of these films was tested after three months. These data, presented in parentheses in Table 4, showed the stability of the composition of the films during this storage time.

#### 3.3.2. FTIR Data

Figure 5 shows the evolution of the FTIR spectra of the obtained samples with a change in the nitrogen flow (F(N_2_) = 0–30 sccm) and substrate temperatures of 25 and 300 °C.

Films synthesized at RT by the MS of the SiC target in an argon medium without nitrogen addition are characterized by the presence in the spectrum (Figure 5a) of the band of Si–C stretching vibration in silicon carbide at 790 cm^–1^ [52]. The addition of even an insignificant amount of nitrogen to the gas phase (F(N_2_) = 2 sccm) leads to a shift of the baseband to the region of large wave numbers and to its broadening, which corresponds to the appearance of bonds between silicon and nitrogen. The observed wide absorption band in the range of 700–1200 cm^−1^ is the typical one for Si–C–N films [1]. It can be interpreted as the superposition of bonds such as Si–C, Si–N, C–N, and Si–O. The bands at 790 and 950 cm^−1^ correspond to Si–C and Si–N stretching vibrations, respectively [52]. Absorptions in the regions of 1055–1400 cm^–1^ and 1020–1100 cm^–1^ are attributed to the vibration of the C–N and Si–O bonds, respectively [1]. A further increase in the nitrogen flow rate causes the appearance in the FTIR spectra and the increase in the intensity of the peaks at 1550–1600 and 2200 cm^–1^. The first broad band is believed to correspond to C=N and/or C=C stretching vibration modes. The vibration frequencies of the doubly bonded groups are very close, so the separation of these groups is quite problematical [1,52]. The band in the 2200 cm^−1^ region corresponds to the stretching vibration of the carbon–nitrogen triple bond C≡N [1]. This fact indicates the formation of a larger number of bonds between nitrogen and carbon atoms with an increase in the nitrogen flow rate. Thus, in general, the inclusion of nitrogen atoms in the film with the rise of F(N_2_) occurs due to the formation of bonds by nitrogen atoms with both silicon and carbon atoms. The evolution of the FTIR spectra of the films obtained at 300 °C is similar (Figure 5b). Similar behavior of the double and triple bonds of carbon and nitrogen in the FTIR spectra of the SiC_x_N_y_ films obtained by magnetron sputtering has been observed in [7,8,27,53].

#### 3.3.3. Raman Spectroscopy

For more information on the composition of the films, Raman spectra were recorded. This technique is employed to extract the structural information and the presence of carbon as an additional phase. The Raman spectra of SiC_x_N_y_ films, obtained at different values of the nitrogen flow rate at room temperature and 300 °C, are shown in Figure 6.

Along with the fluorescent background, four different bands can be distinguished in the spectra. The spectra contain signals from a silicon substrate (520 and 960 cm^–1^) and two broad peaks in the regions of 600–800 and 1100–1700 cm^–1^. Only the spectra of the films deposited using the Ar + N_2_ mixture include the peak around 720 cm^–1^ associated with the Raman oscillation band of C–N bonds [54]. These results are in agreement with those already shown by FTIR spectroscopy. For all samples, the Raman spectra are dominated by the band in the region between 1100 and 1700 cm^–1^ [55]. A similar evolution of the spectra of SiC_x_N_y_ films synthesized in such a system is presented in [9,33]. This corresponds to a specific structure that could be described as a random covalent mixture of sp^2^ and sp^3^ hybridized carbon atoms with distortions of their bond angles and lengths [55]. As can be seen from Figure 5, when nitrogen is added to the gas mixture, the shape of the Raman spectra changes: the baseband broadens and shifts into the region of large wave numbers. The maximum of this band shifts from 1430 to 1480 cm^–1^ and from 1434 to 1487 cm^–1^ for films obtained at room temperature and upon heating, respectively. The baseband can be decomposed considering two contributions: the D and G bands for carbon materials (Appendix A). In amorphous carbon-based materials, the D band becomes Raman active due to the absence of long-range order. The D band arises from limitation in the graphite domain size, induced by grain boundaries or imperfections, such as substitutional N atoms, sp^3^ coordinated carbon, and nitrogen bonded to sp^3^ coordinated carbons. The Raman spectra were fitted with two components in order to determine the D and G band parameters, such as position and intensity (Table 5).

With the increase in the nitrogen flow rate, both the D and G bands shift toward higher wavenumbers. The relative intensity (I_D_/I_G_) and L_a_ increases when the nitrogen content increases. Thus, the addition of nitrogen to the gas phase along with an increase in synthesis temperature leads to the growth of the carbon particles during the sputtering process.

#### 3.3.4. SiC_x_N_y_ Film Chemical Bonding State

The X-ray photoelectron spectroscopy of the core levels allows one to determine the elemental composition of the surface, the concentration of elements on the surface, and the chemical state of atoms on the surface and in near-surface layers. This method was used to study the SiC_x_N_y_ films deposited at 300 °C at different nitrogen flow rates (series C). The XPS data indicate the presence of Si, C, N, and O atoms. Also, a negligible amount of argon was determined. Table 6 presents the atomic ratios and the relative concentrations of the elements in the near-surface layer of the samples. For comparison, there are also EDX data in the table. As could be expected, an increase in F(N_2_) leads to an increase in the nitrogen content in the films. The dependences of the element concentrations obtained by both the XPS and EDX methods have the same trend. It is noted that the XPS data show the higher carbon and oxygen content due to surface hydrocarbon contamination.

Figure 7 displays the evolution of the Si2p, C1s, and N1s core-level spectra with nitrogen flow rate.

The asymmetric shape of the peaks connects with the change in the nature of the chemical environment of the atom. These peaks are decomposed into several components, the assignment of which to certain types of bonds is taken from the literature [1]. The experimental spectra are fitted with a combination of three, four, and two contributions to the Si2p, C1s, and N1s lines, respectively. Table 7 shows the values of the binding energies.

The Si2p spectrum of the film obtained at F(N_2_) = 0 sccm showed three peaks at 100.4, 101.4, and 102.2 eV. The intensity contribution of the individual component resulting from the deconvolution of XPS spectra are given in the brackets in Table 7. The most intense peak (80%) at 100.4 eV corresponds to Si–C bond. Note that the C1s peak arising at 282.8 eV was also corresponded to the C-Si bond. The components at 101.4 and 102.2 eV with lower intensity can be attributed to Si-N and Si-O bonds, respectively. The intensity contribution of the corresponding components in Si2p-region was 5 and 15%. It should be noted, that in this case the contribution of Si−N component is negligible. This fact is in good agreement with the intensity of N-Si component arose at 397.1 eV in the N1s-region.

The addition of nitrogen during the film deposition leads to a shift of the Si2p-peak maximum to the higher energy region. The intensity contribution of Si–C component (~100.4 eV) sharply decreased up to 5%. At the same time, the intensity contribution of the C−Si component was sharply decreased from 40 to 5% in the C1s-region. The Si2p-region exhibits the intense peak (71–75%) at 101.5 eV corresponded to Si-N bonds. Note that component corresponded to N-Si bond at 397.5 eV in the N1s-region with an intensity contribution of ~83–89%.

As can be seen in Figure 7, the C1s peak has a complex structure. A general feature of all three samples is the presence of a dominant peak at 284.5 eV with an intensity contribution of 50–69%. This peak can be attributed to sp^2^-hybridized carbon. Attention should be paid to the behavior of the C−N component. The intensity contribution in C1s-region of the corresponding components increased from 7 to 19% with an increase of F(N_2_). At the same time, the intensity contribution of the N−C component in the N1s-region also increases from 10 to 16%. This result is in a good agreement with the FTIR spectroscopy data discussed above.

The component at 102.2–102.9 eV of Si2p-peak can be attributed to the formation of Si–O bonds. Wherein, the addition of nitrogen leads the shift of Si–O component to the high energy region. This is due to the fact that the electronegativity of nitrogen is larger compared to the carbon. Note that, in case of F(N_2_) = 0 sccm, the silicon atoms are predominantly surrounded with oxygen and/or carbon atoms, and the Si-O component have a lower binding energy (102.2 eV) [56,57]. Whereas at F(N_2_) = 15, 30 sccm, the silicon atoms are fourfold coordinated with oxygen and/or nitrogen atoms and corresponding component have a higher binding energy (102.2 eV) [58]. At the same time, the high energy components at 288.3–288.7 eV in the C1s-region corresponds to carbonate and carboxylic groups. The intensity of these components increased from 3 to 7 % with an increase in the nitrogen flow rate. Thus, one can conclude that the surface of the SiC_x_N_y_ films contained more nitrogen are easily oxidized.

### 3.4. Functional Properties of SiC_x_N_y_ Films

#### 3.4.1. Wettability of SiC_x_N_y_ Films

The wettability of the obtained SiC_x_N_y_ coatings with water was studied as a function of the nitrogen flow rate in the gas mixture. The value of the contact angle was measured in the regime of a sessile drop. The calculations of the observed contact angles carried out according to two algorithms: Ellipse-Fitting and Young–Laplace algorithm are shown in Table 8.

Both calculation results correlate well with on another. All studied samples are hydrophilic. The values of the contact angles of the room temperature films vary in the range of 37 to 50.7°. Strict dependence on the magnitude of the nitrogen flow, i.e., the nitrogen content in the obtained films, is not observed. It should be emphasized that additional studies of the surface of these samples are needed to explain the wettability behavior. The contact angles of films deposited at 300 °C decrease from 67 to 61° with an increase in the nitrogen flow rate; thus, hydrophilization of the surface occurs. These values, taking into account the chemical composition of the films, fit well into the literature data of the contact angles for silicon carbide about 68° [59] and silicon nitride about 60° [60]. According to the FTIR and XPS results adding the nitrogen in the gas phase during deposition leads to decrease in Si–C bond content and increase in Si–N bond content. Also, it was found that the appearance and increase in the content of double and triple CN bonds. Analysis of the near-surface layer of films by XPS showed the presence and slight increase in the content of Si–O and C–O bonds. By increasing both C–N and oxygen-containing bond contents, which have more polarizability, wettability was increased for films.

#### 3.4.2. Optical Properties of Films

The optical performance of SiC_x_N_y_ depends on the conditions under which the films are deposited. Ellipsometry and spectrophotometry were used to study the properties of the SiC_x_N_y_ films deposited at room temperature (series A). Figure 8 shows the dependencies of the refractive index values on the nitrogen flow rate.

The values of n decrease from 2.85 to 2.04 at the 632.8 nm wavelength with an increase in F(N_2_). In this case, a larger number of nitrogen atoms are included in the structure of SiC_x_N_y_ film. The decrease in the refractive index values is associated with an increase in the proportion of Si–N bonds in the films according to EDX, XPS, and FTIR spectroscopy data. As is known, SiC and Si_3_N_4_ have refractive indices equal to 2.6 and 2.0, respectively. In work [18], a similar dependence was found. When RF power decrease to 50 W the n values decreasing to 1.94 (series B).

The optical transmittance of SiC_x_N_y_ film/fused silica structures strongly depends on the film composition and varies from 32 to 60% in the visible region of the spectrum (Figure 8b). As the nitrogen flow rate increases, the transmittance of the samples increases, and the absorption edge shows a blue shift, because the incorporation of nitrogen into SiC_x_N_y_ leads to a smaller effect of the absorbing properties of silicon carbide. The optical band gap E_g_ was estimated in accordance with the Tauc law from the energy dependence of the absorption coefficient (Figure 8c). The value of E_g_ increases from 1.81 to 2.25 eV with increasing nitrogen flow rate (Figure 8d). Reducing the sputtering power to 50 W leads to coatings with a larger band gap equal to 2.53 eV (series B). The smallest band gap is characteristic of films close in composition to SiC. The optical band gap of SiC is reported to lie in the range of 2.3 to 3.3 eV for the most common polytypes of SiC such as 6H-SiC, 4H-SIC, and 3C-SiC [61]. The experimental band gap of the stoichiometric Si_3_N_4_ films is reported to be about 3.3 eV [62]. CVD SiN_x_ films have values for Eg which range between 2.4 and 5.3 eV [63,64].

Our results and their trends are in good agreement with the E_g_ values of SiC_x_N_y_ films obtained by reactive sputtering of a SiC target [65]. It should be noted that the band gap of hydrogenated SiC_x_N_y_:H layers deposited using MS and complex gas mixtures of nitrogen and argon with hydrogen [29], methane [20], and acetylene [19,31] or obtained at a higher synthesis temperature [18], has a higher value and reaches 3.8 eV. In fact, hydrogenated SiC_x_N_y_:H films obtained by plasma enhanced chemical vapor deposition method can exhibit larger values of E_g_ = 4.7 eV [66].

#### 3.4.3. Mechanical Properties

The mechanical properties were evaluated in terms of hardness, elastic modulus, elastic recovery according to the method described in detail earlier [67], and scratch resistance by nanoindentation. For the room temperature films (series A), the values of hardness and elastic modulus decreased as the nitrogen flow rate increased and varied in the ranges of 34.4–16.2 GPa and 269–153 GPa, respectively (Figure 9a,c).

In case of the films deposited at 300 °C (series C), the change in the mechanical properties had a similar trend and varied in the intervals of H = 27.3–17.1 GPa and E = 198–164 GPa (Figure 9b,d). The maximum hardness is observed in the amorphous film with a composition of SiC_1.16_N_0.12_. The data on the hardness, elastic modulus, and elastic recovery of samples are listed in Table 9. Note that the elastic recovery of all samples has high values and reaches the maximum value of 75% also for SiC_1.16_N_0.12_ film.

The adhesion of the SiC_x_N_y_ films to the Si(100) substrate was evaluated by a scratch test with a variable load. Two scratches were made on each sample using Berkovich indenter with edge- and face-forward orientations. The normal load of the indenter increased linearly from 0.1 mN to 100 mN. The topology of the residual scratch on the surface of SiC_x_N_y_ films was studied by AFM. Scratching the samples (Figure 10) leads to the formation of local fractures around the scratch boundary at certain critical loads: L_cr.1_ for the edge-forward orientation and L_cr.2_ for the face-forward orientation (Table 9).

Different attack angles correspond to different deformation modes. The edge-forward orientation is close to plastic extrusion, while the face-forward orientation is close to microcutting.

This is also manifested in the relief characteristics after the tests (formation of piles at the scratch edges as the tip moves in first and second orientation modes). In this case, no significant exfoliation of the films from the substrates was observed. The data of Table 8 show that sample SiC_0.87_N_2.02_, prepared at RT and 30 sccm of nitrogen flow rate, exhibit the strongest resistance to plastic extrusion, while all samples of this series (T_dep_ = 25 °C) are subject to a different type of deformation, microcutting, to almost the same extent. Increasing the synthesis temperature results in coatings with less resistance to both types of impact.

SiC_1.16_N_0.12_ films close in composition to SiC have a maximum hardness value of 34.4 GPa. The obtained hardness values H = 16.2–22.5 GPa for SiC_x_N_y_ with compositions of SiC_1.08_N_1.93_, SiC_0.87_N_2.02_, SiC_1.14_N_1.25_, and SiC_1.01_N_1.79_, are in good agreement with the data from the literature. Our studies have confirmed the previously discovered pattern of a decrease in the value of mechanical characteristics with an increase in the nitrogen content in SiC_x_N_y_ films obtained in similar MS processes.

## 4. Conclusions

In the present work, a series of reactive sputtered SiC_x_N_y_ films were deposited at 25 and 300 °C and 50–150 W of plasma power with an F(N_2_)/[F(N_2_)+F(Ar)] ratio that varied from 0 to 50%. The influence of these variations in the deposition parameters on the films’ growth rates, compositions, chemical bonding, functional properties, and stability was studied. As a result of the optimization of the sputtering process, i.e., using 150 W and F(N_2_) = 10 sccm at RT, growth rate values as high as 103 nm/min were achieved. The SEM and AFM results showed that adding nitrogen had no considerable effect on the morphology and roughness of the SiC_x_N_y_ films. The elemental composition of the films was found to be strongly affected by the nitrogen-argon flow ratio. The EDX results showed that, by increasing the nitrogen flow rate up to 30 sccm, the nitrogen content in the films’ structure rose, and the film composition varied from SiC_1.33_N_0.04_ to SiC_0.87_N_2.02_. The study of the aging of the SiC_x_N_y_ films over the course of five months showed the stability of their elemental composition in the ambient air under the standard conditions. In the studied range of conditions, all of the synthesized SiC_x_N_y_ films were found to be amorphous, with embedded carbon particles. The Raman results indicated that increasing the incorporation of nitrogen in films’ structure increased the size of the particles. The FTIR results showed a decrease in the Si–C bond content and an increase in the Si–N bond content after adding the nitrogen in the films’ structure. Also, an appearance and increase in the content of double and triple CN bonds was found. This trend was confirmed by the XPS data. An analysis of the near-surface layer of the films showed the presence of and slight increase in the content of Si–O and C–O bonds.

The functional properties, such as the refractive index, transmittance, optical band gap, hardness, Young’s modulus, elastic recovery, critical loads, and the contact angle can be tuned by adjusting the MS process parameters. The refractive index can be tuned from 2.85 to 2.04, with an increase in F(N_2_). The band gap values ranged from 1.81 to 2.53 eV. The highest values of hardness (34.4 GPa) and Young’s modulus (269 GPa) were measured for SiC_1.16_N_0.12_ film deposited at room temperature. The highest resistance to scratching was also demonstrated by the films deposited at room temperature. The deposition parameters for producing of the hydrophilic films with the minimal value of CA equal to 37° were found. Thus, the produced SiC_x_N_y_ films can be considered as a promising coating for biomedical and optical applications.

## Figures and Tables

**Figure 1 materials-16-01467-f001:**
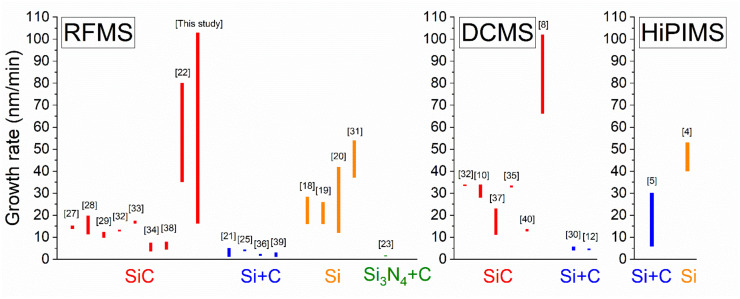
Growth rate dependence on MS mode and type of sputtered target. Hänninen T., et al., 2018 [4]; Pettersson M., et al., 2013 [5]; Kozak A.O., et al., 2017 [8]; Kulikovsky V., et al., 2014 [10]; Berlind T., et al., 2001 [12]; Bachar A., et al., 2018 [18]; Peng Y., et al., 2011 [19]; Saito N., et al., 1991 [20]; Li Q., et al., 2017 [21]; Sundaram K.B., et al., 2020 [22]; Li Q.et al., 2021 [23]; Hüger, E., et al., 2012 [25]; Peng Y., et al., 2018 [27]; Peng Y., et al., 2018 [28]; Peng Y., et al., 2014 [29]; Medeiros H.S., et al., 2011 [30]; Peng, Y., et al., 2010 [31]; Bhattacharyya A.S., et al., 2009 [32]; Bhattacharyya A.S., et al., 2009 [33]; Fraga M.A., et al., 2008 [34]; Mishra S.K., et al., 2007 [35]; Du X.-W., et al., 2007 [36]; Wei J., et al., 2000 [37]; Xiao X., et al., 2000 [38]; Lutz H, et al., 1998 [39]; Scharf T.W., et al., 1997 [40].

**Figure 2 materials-16-01467-f002:**
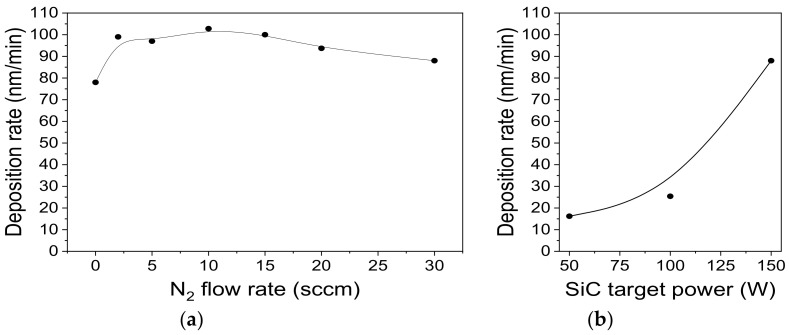
(**a**) Deposition rate of films versus nitrogen flow rate (series A); (**b**) Film growth rate as a function SiC target power (series B).

**Figure 3 materials-16-01467-f003:**
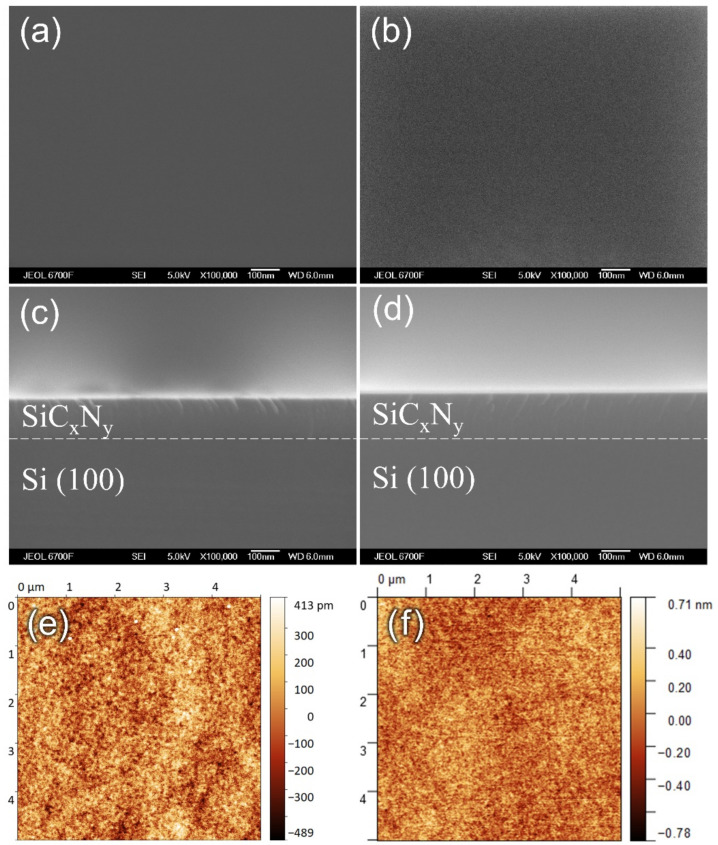
SEM images: top view of SiC_x_N_y_ film prepared at RT and (**a**) 0, (**b**) 30 sccm of nitrogen flow rate; cross-section view for (**c**) 0, (**d**) 30 sccm and AFM images for (**e**) 0, (**f**) 30 sccm flow rate (series A).

**Figure 4 materials-16-01467-f004:**
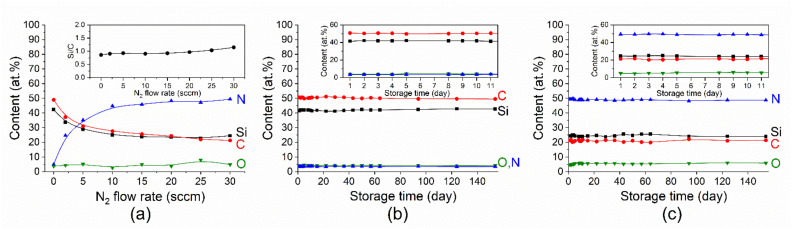
(**a**) The dependence of the elemental composition and the ratio of Si/C elements (insert) in SiC_x_N_y_ films on the nitrogen flow obtained at room temperature (series A). The element contents in SiC_x_N_y_ films obtained at (**b**) F(N_2_) = 0 and (**c**) 30 sccm over time during five months. The insets in (**b**,**c**) shows the elemental composition during the first storage days.

**Figure 5 materials-16-01467-f005:**
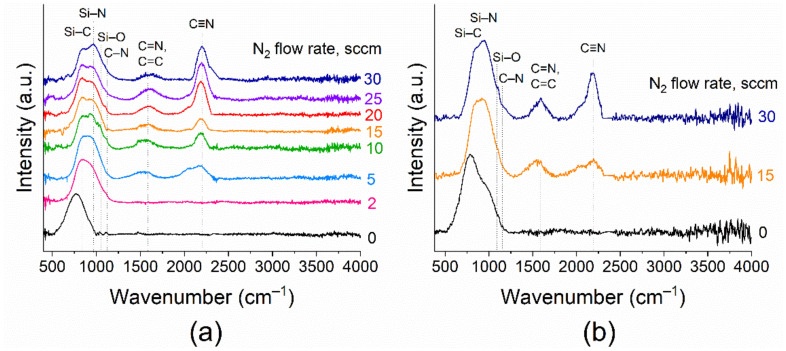
FTIR spectra evolution of SiC_x_N_y_ films obtained by sputtering at (**a**) room temperature and (**b**) 300 °C with the different nitrogen flow (series A and C).

**Figure 6 materials-16-01467-f006:**
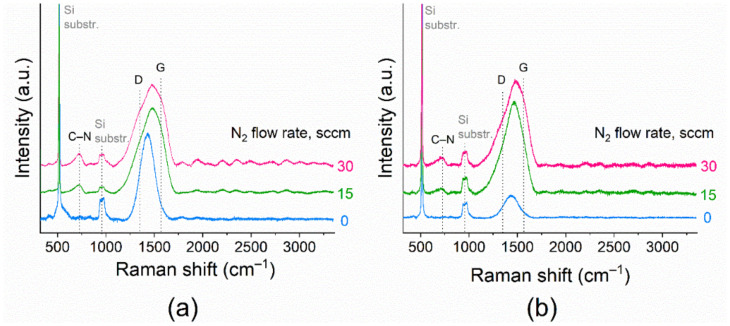
Raman spectra of SiC_x_N_y_ films, deposited at (**a**) T_dep_ = 25 and (**b**) 300 °C (series A and C).

**Figure 7 materials-16-01467-f007:**
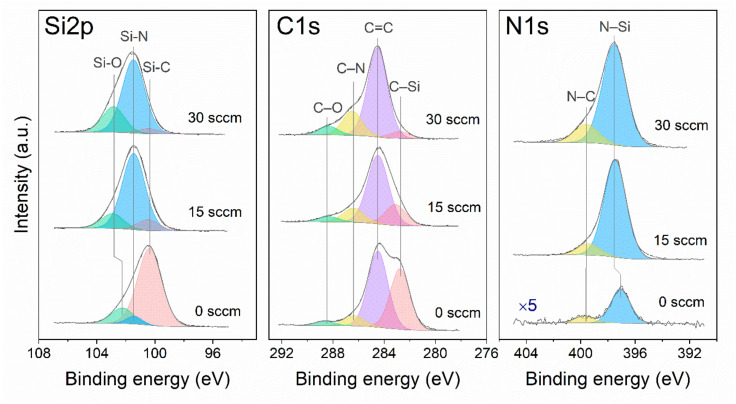
XPS spectra of SiC_x_N_y_ films deposited at T_dep_ = 300 °C and different nitrogen flow rates.

**Figure 8 materials-16-01467-f008:**
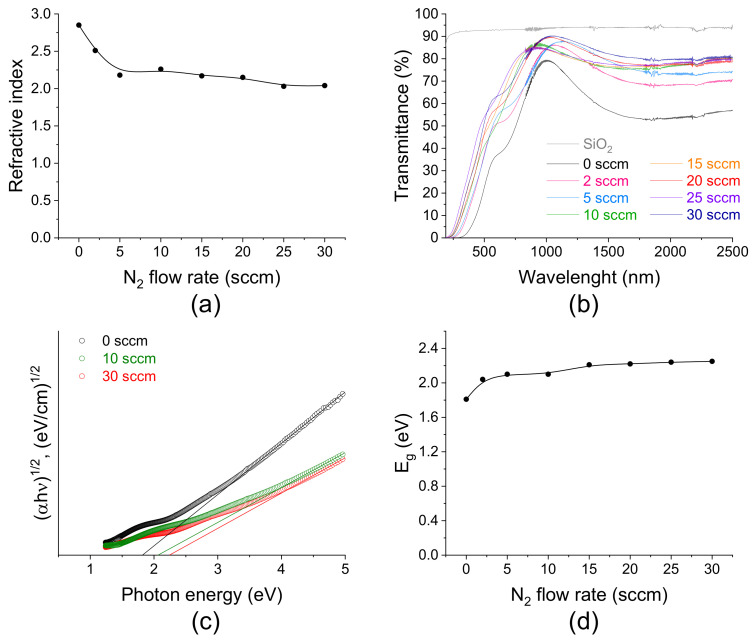
Optical properties of SiC_x_N_y_ films deposited at various nitrogen flow rates (series A): (**a**) refractive index, (**b**) transmittance, (**c**) Tauc plots versus photon energy with linear approximation and (**d**) band gap energy.

**Figure 9 materials-16-01467-f009:**
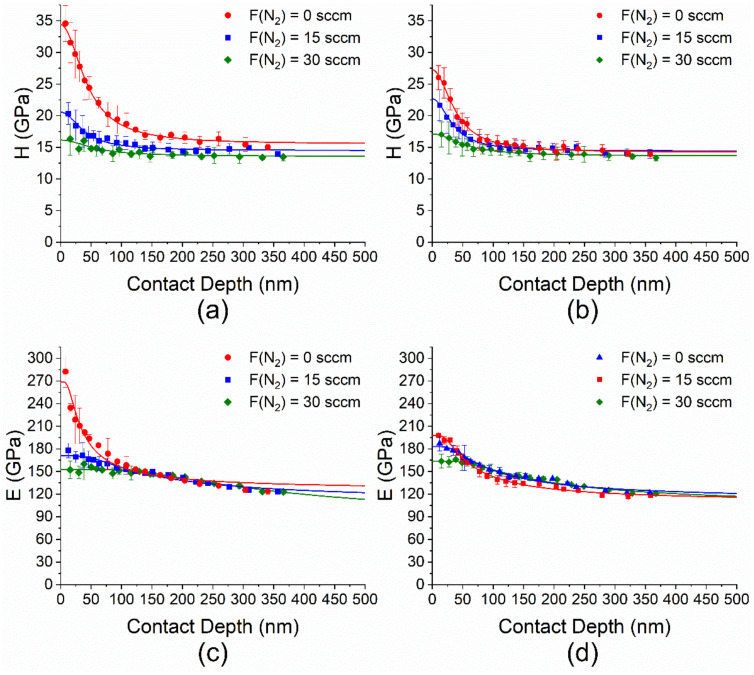
Hardness (**a**,**b**) and elastic modulus (**c**,**d**) of SiC_x_N_y_ samples as functions of the indenter penetration depth at the different nitrogen flow rate. T_dep_ = RT (**a**,**c**) and 300 °C (**b**,**d**).

**Figure 10 materials-16-01467-f010:**
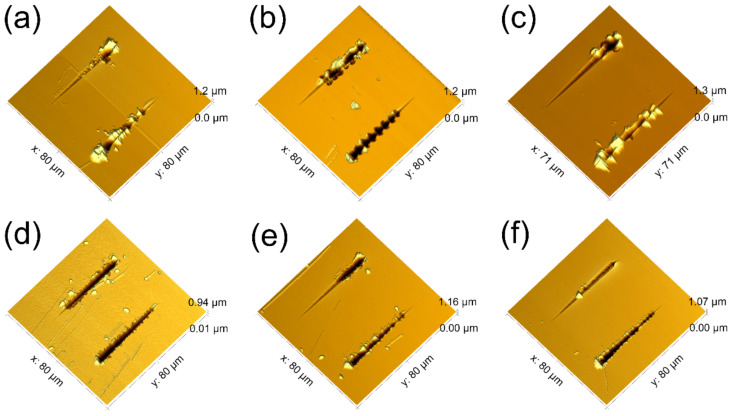
AFM images of scratches on the surface of SiC_x_N_y_ films obtained at (**a**,**b**,**c**) room temperature and (**d**,**e**,**f**) 300 °C and nitrogen flow rate of (**a**,**d**) 0, (**b**,**e**) 15 and (**c**,**f**) 30 sccm.

**Table 1 materials-16-01467-t001:** Deposition parameters of PVD SiC_x_N_y_ films and their functional characteristics.

Deposition Conditions				Film Properties		Functional Characteristics			Ref.
Targets	Additional Gases	Power, W	T_dep_, °C	d_film_, µm	RMS, nm	Optical	Mechanical	Wettability	
						n (λ, nm) T, % (range, nm) E_g_, eV	H, GPa E, GPa R, %	CA, °	
HiPIMS									
Si	C_2_H_2_, N_2_, Ar	1200	RT–340	0.3–0.9	–	–	5–23 50–190 –	–	[4]
Si; C	N_2_, Ar	1000–1400; 0–1400	110, 430	0.5–0.9	0.2–3.8	–	10–21 140–220 –	–	[5]
Si; C	N_2_, Ar	1000–1400; 0–1400	110, 430	0.4–0.9	–	–	10–21 140–220 –	–	[6]
DCMS									
SiC	N_2_, Ar	320	RT	2.2–2.7	1	–	18–23 242–293 –	–	[7]
SiC	N_2_, Ar	–	350	0.6–0.9	0.23, 0.28	–	16–22.5 165–205 –	–	[8]
SiC	N_2_, Ar	320	RT–600	2.2–3.4	–	–	18.5–22.5 270–285 –	–	[9]
SiC	N_2_, Ar	320	RT	2.2–2.7	–	–	10–23 230–295 –	–	[10]
Si; C	N_2_, Ar	– –	500	1.5	–	–	16.5–29 – –	–	[11]
Si; C	N_2_, Ar	17.5; 100	100 350	0.5	2	–	9–28 – –	57–87	[12]
RFMS									
Si	CH_4_, N_2_, Ar	300	400	0.375–0.614	–	1.9–2.5 (633) – 2.0–3.8	– – –	–	[18]
Si	C_2_H_2_, N_2_, Ar	–	RT	–	–	– 40–95 (400–1000) 2.1–2.9	– – –	–	[19]
Si	CH_4_, N_2_, Ar	200	300	–	–	– – 1.9–2.4	– – –	–	[20]
Si; C	N_2_, Ar	210–270; 80–110		0.099–0.459	–	– 97–89 (200–1200) 3.64–3.98	– – –	–	[21]
SiC	N_2_, Ar	400	500	4	–	– – –	20 210 45	–	[13]
SiC	N_2_, Ar	100	–	–	65–15	– – –	5.8–12.2 73.5–117.8 –	–	[14]
SiC	N_2_, Ar	100	RT	–	–	– – –	13–17.5 – –	–	[15]
SiC	N_2_, Ar	200–400	RT–600	3.5–4.1	–	– – –	8–22 – –		[16]
SiC	N_2_, Ar	100	–	–	–	– 95 (700–800) 2.28 – 2.77	– – –	–	[22]
SiC; Si_3_N_4_	N_2_, Ar; CH_4_, Ar	220–600 140–600	RT	–	–	– – –	12–15.5 95–135 –	–	[17]
Si_3_N_4_; C	Ar	100–190 80–110	RT, 500–700	0.12	2.2	– 2–70 (400–800) 3.0–5.1	– – –	–	[23]
SiC	N_2_, Ar	50–150	RT, 300	0.14–0.32	0.2	1.94–2.85 (632.8) 32–60 (550) 1.81–2.53	16.2–34.4 153–269 64–75	37–67	This study

T_dep_ —deposition temperature; d_film_—film thickness; RMS—root mean square roughness; n—refractive index; λ—wavelength; T—range of transmittance value; E_g_—optical band gap; H—hardness; E—Young’s modulus; R—elastic recovery; CA—contact angle; HiPIMS—high-power impulse magnetron sputtering; DCMS—direct current magnetron sputtering; RFMS—radio frequency magnetron sputtering; RT—room temperature.

**Table 2 materials-16-01467-t002:** Mechanical properties and composition of SiC_x_N_y_ films obtained by magnetron sputtering.

Deposition Technique	H, GPa	E, GPa	Elemental Composition						Chemical Bonds	Ref.
			Si	C	N	O	H	Ar		
RFMS DCMS HiPIMS	22 21 19	225 223 202	50 36 23	25 29 25	25 35 52	– – –	– – –	– – –	C≡N, Si−C, Si−N, C=N/C−C, C−N, C−H, N−H Si−C, Si−N, C≡N, C=N/C−C, C−N Si−C, Si−N, C≡N, C=N/C−C, C−N	[41]
RFMS	28	226	47	25	25		3		Si−C, Si−N, C–C/C–H, Si–O	[42]
RFMS DCMS	22 16	240 200	32 20	39 56	29 44	– –	– –	– –	− –	[32]
DCMS	22	–	41	45	14	–	–	–	Si−N, C−N, C≡N, Si−C−N	[7]
DCMS	19	185	34	39	20	7	–	–	Si−C, Si−N, C−N, Si−O, C≡N	[8]
DCMS	22	–	41	45	14	–	–	–	−	[9]
HiPIMS	22	185	40	8	38	3.7	8	2.3	Si−C, Si−N, Si−O, C−C, C−N, C−O	[4]

DCMS—direct current magnetron sputtering; RFMS—radio frequency magnetron sputtering; HiPIMS—high-power impulse magnetron sputtering.

**Table 3 materials-16-01467-t003:** Experimental conditions and chemical composition of MS SiC_x_N_y_ films.

Series	T_dep_, °C	RF Power, W	F(N_2_), sccm	F(Ar), sccm	F(N_2_)/[F(N_2_) + F(Ar)]	Deposition Time, min	d, nm	Composition of SiC_x_N_y_
A	25	150	0	60	0.00	3.0	233	SiC_1.16_N_0.12_
			2	58	0.03	3.0	297	SiC_1.10_N_0.73_
			5	55	0.08	3.0	291	SiC_1.08_N_1.22_
			10	50	0.17	2.5	257	SiC_1.10_N_1.79_
			15	45	0.25	2.5	250	SiC_1.08_N_1.93_
			20	40	0.33	3.0	281	SiC_1.03_N_2.04_
			25	35	0.42	3.0	240	SiC_0.96_N_2.07_
			30	30	0.50	3.0	265	SiC_0.87_N_2.02_
B	25	50	30	30	0.50	9.0	146	SiC_1.19_N_1.97_
		100				6.5	165	SiC_1.10_N_1.76_
		150				3.0	265	SiC_0.87_N_2.02_
C	300	150	0	60	0.00	3.0	140	SiC_1.33_N_0.04_
			15	45	0.25	2.5	160	SiC_1.14_N_1.25_
			30	30	0.50	2.4	170	SiC_1.01_N_1.79_

T_dep_—deposition temperature; RF—radio frequency; d—film thickness.

**Table 4 materials-16-01467-t004:** Elemental composition of SiC_x_N_y_ films obtained at a substrate temperature of 300 °C.

F(N_2_), sccm	Si, at.%	C, at.%	N, at.%	O, at.%	Si/C
0	36.0 (35.4) *	47.7 (47.0)	1.4 (1.9)	14.9 (15.7)	0.8
15	26.7 (27.0)	30.4 (30.8)	33.3 (32.4)	9.6 (9.8)	0.9
30	23.7 (23.7)	23.9 (25.6)	42.4 (40.0)	10.0 (10.6)	1.0

* Note: the elemental concentrations in the samples after three months of air storage are given in the parentheses.

**Table 5 materials-16-01467-t005:** Position of D and G band and ratio of their relative intensity for SiC_x_N_y_ films versus the nitrogen flow ratio at RT and 300 °C deposition temperature.

T_dep_, °C	F(N_2_), sccm	D Peak, cm^–1^	G Peak, cm^–1^	I_D_/I_G_	L_a_, nm
25	0	1385	1470	1.24	1.50
	15	1391	1549	1.52	1.66
	30	1397	1571	2.25	2.02
300	0	1400	1478	1.47	1.63
	15	1438	1540	1.83	1.82
	30	1453	1570	2.51	2.14

**Table 6 materials-16-01467-t006:** Atomic ratios of elements and elemental composition of SiC_x_N_y_ films * from the XPS measurement. The EDX data are given in parentheses.

F(N_2_), sccm	[C]/[Si]	[N]/[Si]	[O]/[Si]	Elemental Composition			
				Si, at.%	C, at.%	N, at.%	O, at.%
0	1.7	0.05	0.75	28.6 (36.0)	48.6 (47.7)	1.5 (1.4)	21.4 (14.9)
15	1.3	0.79	0.81	25.6 (26.7)	33.3 (30.4)	20.3 (33.3)	20.8 (9.6)
30	1.4	0.95	0.98	23.1 (23.7)	32.3 (23.9)	21.9 (42.4)	22.6 (10.0)

* Note: the substrate temperature during the films deposition was of 300 °C.

**Table 7 materials-16-01467-t007:** XPS analyzed binding energy for Si2p, C1s, and N1s core levels.

F(N_2_), sccm	Si2p			C1s				N1s	
	Si–C	Si–N	Si–O	C–Si	C–C	C–N	C–O	N–Si	N–C
0	100.4 (80%)	101.4 (5%)	102.2 (15%)	282.8 (40%)	284.5 (50%)	286.4 (7%)	288.7 (3%)	397.1 (90%)	399.8 (10%)
15	100.5 (11%)	101.5 (75%)	102.9 (14%)	283.1 (19%)	284.5 (63%)	286.4 (13%)	288.3 (5%)	397.5 (89%)	399.4 (11%)
30	100.4 (5%)	101.5 (71%)	102.9 (24%)	282.8 (5%)	284.5 (69%)	286.5 (19%)	288.3 (7%)	397.5 (84%)	399.6 (16%)

**Table 8 materials-16-01467-t008:** The values of the contact angles of SiC_x_N_y_ films (series A and C).

T_dep._, °C	N_2_ Flow Rate, sccm	CA, °	
		Ellipse-Fitting	Young–Laplace
25	0	39 ± 2	39 ± 2
	2	37 ± 1	37 ± 1
	5	44 ± 3	44 ± 3
	10	49 ± 3	48 ± 1
	15	49.3 ± 0.7	50.7 ± 0.7
	20	48.8 ± 0.6	48.7 ± 0.7
	25	41.1 ± 0.5	41.1 ± 0.3
	30	42 ± 2	43 ± 2
300	0	67 ± 4	67 ± 4
	15	61 ± 8	61 ± 8
	30	60 ± 1	61 ± 2

T_dep_−deposition temperature; CA−contact angle.

**Table 9 materials-16-01467-t009:** Mechanical properties of SiC_x_N_y_ films using nanoindentation measurements (series A and C).

T_dep_, °C	F(N_2_), sccm	H, GPa	E, GPa	R, %	L_cr.1_, mN	L_cr.2_, mN
25	0	34.4 ± 3.4	269 ± 15	75	73	28
	15	20.6 ± 2.2	171 ± 12	68	43	28
	30	16.2 ± 2.7	153 ± 9	64	85	25
300	0	27.3 ± 2.4	198 ± 10	73	25	13
	15	22.6 ± 2.9	183 ± 6	71	62	17
	30	17.1 ± 3.1	164 ± 13	64	41	14

T_dep_−deposition temperature; F(N_2_)−nitrogen flow rate; H−hardness; E−elastic modulus; R−elastic recovery; L_cr.1_−critical load for the edge-forwarded indenter; L_cr.2_−critical load for the face-forwarded indenter.

## Data Availability

Not applicable.

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
