# Peer review of "Enhanced Wettability, Hardness, and Tunable Optical Properties of SiC_x_N_y_ Coatings Formed by Reactive Magnetron Sputtering"

_materials, 2023, doi:10.3390/ma16041467_

Round 1
Reviewer 1 Report
I read the paper and make some observations:
You state in line 290-292 that you did not control the total flow of argon with nitrogen, which certainly affects the deposition pressure. Argon should have been kept constant, because you did not investigate its influence. Try to justify it.
Line 312 mentions Figure S1, which you did not show in the supplementary materials.
Table 4. Why nitrogen has increased from 1.4 to 1.9 after 5 months. Reason? I'm talking about the first row in the table, where the flow is 0 sccm.
Table 4 and Figure 4. Why are one measurement after 3 months and the other after 5 months, and a comparison is made.
Line 448-449 Not very significant increase in bond energy with increase in F(N2).
Reviewer 2 Report
Manuscript Number: materials-2130972
Title: Enhanced Wettability, Hardness and Tunable Optical Properties of SiCxNy Coatings Formed by Reactive Magnetron Sputtering
The authors present a study concerning Si-C-N films. Globally the results are interesting and valuable and could deserve publication. However, several questions remains unclear.
Consequently, I major revision.
General remarks
1. Minor remark. In introduction and conclusion, I suggest writing “5 months” in full letters: “five months”.
2. Some sentences suffer writing errors. I will point some of them but suggest a deep proof reding before the next submission.
Detailed remarks:
Introduction
1. L64: “targets in these techniques.” I suggest replacing “techniques” by “works”. Or “studies”… however it is used in the following sentence. Eventually, try to change a little the writing of both sentences.
2. L95: “It should be emphasized that in most cases the SiCxNy films have a very low roughness”. This is consistent with most of the films with an amorphous structure. Maybe you could discuss it in one or two small sentences?
3. L97-98: “high-temperature protective applications” could you provide some precise examples of these potential applications?
Same remark for L165: “their potential application in various fields of technology”.
4. L128-129: “the hardness of MS SiCxNy films obtained in RF and DC modes showed that RF sputtering leads to the production of harder coatings”. Ok for this two references, however, in the table 2 of the present work the number of data is too low to conclude definitively. I agree that DCMS films can present lower hardness, but the maximum hardness is found with every technique. Please adapt your sentence.
5. L140: “hard films have different elemental and phase compositions and different chemical structures”à meaning that hardness is not directly driven bay these two criteria and that other phenomenon play a role.
6. L142-144: please add some comments about the possible reason of this increase.
Experimental
7. L182-183: “The substrates were then immediately placed inside the vacuum chamber and evacuated.” The chamber is evacuated (or other better verb) not the substrates. Please correct the sentence.
8. L184: remove “the” after “The SiC (99.9 % purity)”.
9. L189: “target-substrate distance was 30 mm” such a short distance! Does it means that you made one deposition par substrate? what about the thickness uniformity? This can explain the high deposition rates found is this work, as well as the very low roughness. Could you discuss it at their respective sections? It is strange that you do not find more argon in yours films; the ion bombardment must be quite high, even without biasing?
10. L191-194: working pressure. Did the pressure was maintained constant with a throttle valve? With a constant gas flow, if you replace Ar by a reactive gas the pressure should decrease due to the consumption of the nitrogen by the forming film. Could you precise this point?
11. Something is not so logical in table3 (or in the associated text). In the text, you present the series in the order 1, 2, 3 while in the table it is 1, 3, 2. Please, made it consistent. In addition, it could be useful to precise in the table the series number.
12. The deposition time is not specified. As you use it in part 3.1, this should be added to table 3.
13. “Film Stability”. Interesting idea. Why did you not use the same process for the 300 °C series as the RT ones? I suppose that you made the 300 °C after obtaining the results of the RT, however you have to justify this difference? Why repeating this information at the end of part 2.4?
Results and Discussion
14. L283-284: “This phenomenon can be observed due to the formation of a chemical compound” I supposed that you wanted to write “explained” instead of “observed”.
15. L286-287 “The thickness of this layer increases as nitrogen flow rate increases.” Not really. The main criteria is the fraction of target surface (especially the racetrack) covered by the compound layer.
16. L291-292: “which could lead to a change in total pressure in the reaction chamber” strange remark. Previously the pressure was given with a unique value assuming that the pressure was constant. Is it yet not the case? Please check, explain and correct.
Moreover, even with a constant pressure, if you reduce the amount of Ar, you reduce the amount of sputtering ions that induce a decrease of the metallic atoms sputtered and consequently the deposition rate. N has a lower sputtering power than Ar (and the number of N ions create is certainly lower than the Ar ions loss).
17. L293-294 “With W increasing from 50 to 150 W, the growth rate of the films increases” this is not a great finding… of course that if you increase the power the number and energy of the ions will increase and consequently the deposition rate. I you want to keep that part, you should only present it as expected results and not as a discover, even confirmed by the literature.
18. L295-296: “As can be seen from Figure 1, the obtained film deposition rates are higher than previously reported in the literature” what would be really interesting (if you want to keep this sentence) would be to explain why… some clues where given previously in the review.
19. Fig 3. What about the SEM and AFM views of the 300 series? Is the answer in the last sentence of this section? If yes, please introduce before the “no-change” results for the deposition at 300°C.
20. L322: “does not exceed 1 at.%” Obviously on fig 4, the O amount is higher than 1 %... maybe an typing error occurred? I think that 10 % is the correct limit.
21. Figure 4. b and c. I suggest to present the graphs function of the time (month for example, or a kind of non linear scale)
22. L341: “deposited films oxidize once they are in contact with air.” forget it…. If you recognize that you have nitrogen from the residual atmosphere (actually, it is a very small leak… certainly combined with gas desorption due to the high temperature) you have also to assume that the oxygen comes from the in situ process (which is the correct answer…), and not ex situ. You aging tests are proofs of that…
23. L416: “Table 4” should be numbered 6.
24. L466-472: ok the films at 300 °C fits with the literature, however, how do you explain the CA values for the RT series?
25. Figure 8 c. you claim that the Eg values fits with the literature. Why not. However, when I see the picture I have the feeling that the values could be exactly the same if you change a little the slope… could you check and discuss that?
26. Figure 9 to be consistent, please use the same scaling for the contact depth (fig b is different).
27. Figure 10. I do not see what bring this pseudo 3D view. You should pour typical top view (like in figure 3) and use the same z-scale.
Reviewer 3 Report
The paper entitled "Enhanced Wettability, Hardness and Tunable Optical Properties of SiCxNy Coatings Formed by Reactive Magnetron Sputtering" can be published in Materials after the authors clarify the novelty of the work. What they need to succeed? What is the purpose of the particular research work? Can these materials be utilized in a particular application? Please clarify this point in Introduction and Conclusion.
I believe that the Introduction is long and difficult to follow.
Reviewer 4 Report
The XPS analysis and reporting should be improved.
* The data in Figure 7 is presented after background subtraction. The original data should be shown.
* The information about the peak-fitting analysis, including background modeling, is lacking.
* There is no information about how the atomic percentages were calculated. If vendor-provided sensitivity factors were employed, a detailed description of the materials employed to assess them should be provided.
There is a large disagreement between the composition assessed from XPS and EDX. This should be discussed at length.
Round 2
Reviewer 2 Report
Manuscript Number: materials-2130972 R1
Title: Enhanced Wettability, Hardness and Tunable Optical Properties of SiCxNy Coatings Formed by Reactive Magnetron Sputtering
The authors present their revised paper. They answered most of the previous remarks and made the corresponding changes. However, I have still few remarks. Something must be clear: when I ask questions, it is not because I want personally the answer… it is because it is missing in the paper and consequently something has to be corrected in the paper (add the missing information, change an unclear sentence, delete an incorrect information, etc.).
Consequently, I suggest minor revision.
Detailed remarks:
Experimental
1. L189: “target-substrate distance was 30 mm” such a short distance! Does it means that you made one deposition par substrate? what about the thickness uniformity? This can explain the high deposition rates found is this work, as well as the very low roughness. Could you discuss it at their respective sections? It is strange that you do not find more argon in yours films; the ion bombardment must be quite high, even without biasing?
2. L215: I’m sorry, I have miss this question previously: the target diameter is missing? Please add it.
3. L221: working pressure. Thank you for the answer. However, this information must be put in the paper. Please add it.
4. Table3. If there is no important reason to order the series in the table, please made it consistent with the text… i.e. RT at 150 W, 300 °C at 150W and finally RT at various power.
5. Table 3. Thank you for adding the deposition rate range. However, the detailed information must be put in table 3. Please add it.
6. “Film Stability”. Thank you for the answer. However, this information must be put in the paper. Please add it.
Results and Discussion
7. L286-287 “The thickness of this layer increases as nitrogen flow rate increases.” I agree with the Japanese colleagues. However, the main criteria is the fraction of target surface (especially the racetrack) covered by the compound layer and not the thickness. Keep the mention with the increasing thickness but add the increase of the poisoned fraction surface.
8. L377: “does not exceed 1 at.%”. indeed… it is argon… I was focused on the oxygen. Then, please add the average amount L375.
9. L405: “deposited films oxidize once they are in contact with air.” As I say previously… forget it… oxidation is in situ and not ex situ. The aging tests are proofs of that… if there is a small oxidation at the first contact with air, it will be few nm at the top surface. Eventually, you can see that with XPS but not with EDX. The amount of oxygen measured with EDX comes from the inside of the film and consequently where incorporated during the deposition. Please delete the last sentence of the paragraph.
10. L551-557: CA. Thank you for the answer about the RT films. However, this information must be put in the paper. There is no shame to have still investigation to perform… Please add it.
Reviewer 3 Report
I have no further comments.
Reviewer 4 Report
The concerns about the XPS data analysis were not addressed. The small changes on the text do not solve the problem.
